# Variations in the Human Serum Albumin Gene: Molecular and Functional Aspects

**DOI:** 10.3390/ijms23031159

**Published:** 2022-01-21

**Authors:** Gianluca Caridi, Francesca Lugani, Andrea Angeletti, Monica Campagnoli, Monica Galliano, Lorenzo Minchiotti

**Affiliations:** 1UOC Nefrologia e Trapianto Rene, Laboratorio di Nefrologia Molecolare, Istituto Giannina Gaslini IRCCS, 16147 Genova, Italy; francescalugani@gaslini.org (F.L.); andreaangeletti@gaslini.org (A.A.); 2Department of Molecular Medicine, University of Pavia, 27100 Pavia, Italy; monica.campagnoli@unipv.it (M.C.); galliano@unipv.it (M.G.)

**Keywords:** genetic variants, structural changes, stability: ligand binding, pharmacokinetics, congenital analbuminaemia

## Abstract

The human albumin gene, the most abundant serum protein, is located in the long arm of chromosome 4, near the centromere, position 4q11–3. It is divided by 14 intervening introns into 15 exons, the last of which is untranslated. To date, 74 nucleotide substitutions (mainly missense) have been reported, determining the circulating variants of albumin or pre-albumin. In a heterozygous state, this condition is known as alloalbuminaemia or bisalbuminaemia (OMIM # 103600). The genetic variants are not associated with disease, neither in the heterozygous nor in the homozygous form. Only the variants resulting in familial dysalbuminaemic hyperthyroxinaemia and hypertriiodothyroninaemia are of clinical relevance because affected individuals are at risk of inappropriate treatment or may have adverse drug effects. In 28 other cases, the pathogenic variants (mainly affecting splicing, nonsense, and deletions), mostly in the homozygous form, cause a premature stop in the synthesis of the protein and lead to the condition known as congenital analbuminaemia. In this review, we will summarize the current knowledge of genetic and molecular aspects, functional consequences and potential therapeutic uses of the variants. We will also discuss the molecular defects resulting in congenital analbuminaemia, as well as the biochemical and clinical features of this rare condition

## 1. Introduction

Human serum albumin (ALB; OMIM # 103600) is constantly released by liver hepatocytes in an amount of 14gr per day in healthy adult subjects and has a half-life of approximately 19 days [1]. ALB is the most abundant protein (approximately 35–50 gr/L) in plasma, representing 60–65% of total proteins, and contributes to approximately 80% of the oncotic plasma pressure [1]. ALB has an unequal content of acidic (98 Glu + Asp) and basic (83 Lys + Arg) residues, resulting in a net charge of −15 at physiological pH, which makes it not permeable to the capillary walls (Donnan effect) [1]. The ability of ALB to bind ligands is largely known; it acts as a fundamental transporter for numerous endogenous and exogenous compounds, in particular for the less soluble and hydrophobic ones [2]. Drugs represent a significative component of the exogenous substances binding ALB. Moreover, ALB is the most abundant circulating antioxidant agent, possessing esterase-like enzymatic activity [3] and heme-based catalytic properties [4,5,6,7]. Several studies suggested that ALB also exerts an anticoagulant effect by binding antithrombin [8] and by inhibiting platelet aggregation [9]. Recent evidence also supports a role in human innate immunity [10].

The mature ALB molecule, with a molecular mass of 66.5 kDa, is a single polypeptide chain of 585 amino acids without prosthetic groups or covalently bound lipids or carbohydrates [1]. The three-dimensional structure of the protein [11,12] is now defined at a resolution of 2.3 Å [2]. The crystallographic studies revealed that ALB has 67% α-helix content and no β-sheet secondary structure, folded into a heart-shaped molecule comprising three homologous domains (I–III). Each domain is composed of two subdomains (A and B), with different helical folding patterns linked by flexible loops. The overall conformations in neutral solution and in crystal structures are very similar [13,14].

ALB belongs to the albumin super family, including, among others, α-fetoprotein, vitamin D-binding protein (Gc-globulin), and afamin [15]. α-Fetoprotein is fundamental during fetal life, but it is not expressed in adulthood. The fifth gene of this super-family, the α-fetoprotein-related gene, has been found in primates as well, but in humans it has several pathogenic variants which turn it into an inactive pseudogene [16]. Moreover, recent phylogenetic and structural analyses revealed that the extracellular matrix protein 1 (ECM1) also belongs to this super-family [17]. The single-copy genes of ALB, vitamin D-binding protein, afamin, α-fetoprotein, and the α-fetoprotein-related gene, are all expressed in a co-dominant manner and are localized in chromosome 4 at position 4q13.3, near the centromere [18]. The five genes have arisen from a common ancestor through a series of duplication events and are tightly linked in humans and in all other species studied [18]. The relation between these genes and the one for ECM1, which is located in a completely different position (1q21.3) [18] has not yet been clarified.

The gene for ALB (*ALB*; NCBI Genomic Sequence: NC_00004.12) is 16,961 nucleotides long from the putative “cap” site to the first poly(A) addition site and is composed of 14 intervening introns and 15 exons, the last of which is untranslated [19]. The exons are symmetrically placed within the three domains of the protein, probably as consequence of the triplication of a single primordial domain [19]. *ALB* is expressed as single-copy gene in a co-dominant manner and both alleles are translated.

The messenger ribonucleic acid (mRNA) of *ALB* (NCBI Reference Sequence NM_000477.7) encodes for a precursor protein named pre-pro-albumin, which consists of 609 amino acids (NCBI Reference Sequence: NP_000468.1), resulting in the mature protein of 585 residues after the further cleavage of 24 residues [1].

ALB is characterized by a significant DNA polymorphism, similarly to the other genes of the albumin super-family, resulting in phenotypes ranging from the presence of genetic variants (alloalbuminaemia or bisalbuminaemia; OMIM # 103600) to the complete absence of the circulating protein (congenital analbuminaemia, CAA; OMIM # 616000). (Figure 1).

## 2. Alloalbuminaemia

Seventy-four different mutations, resulting in seventy-one distinct genetic variants of ALB and pro-albumin have been characterized at the protein and/or gene level. In Table 1, we summarize the list of the variants, together with their main structural and functional properties. A more detailed presentation is reported in a dedicated website [20], managed and continuously updated by our group.

### 2.1. Frequency and Detection of Alloalbumins

Starting from the early 1980s, an extensive survey performed by the Italian Committee for the Standardization of Electrophoretic Laboratory Methods (CISMEL) allowed for the identification of the Italian ALB variants. Sera derived from patients, blood donors, or from a large cohort of healthy individuals recruited for genetic studies, were screened by a cellulose acetate electrophoretic at pH 8.6 and 5.0 [15], allowing for the identification of several alloalbumins [15]. In such electrophoretic analysis, albumins from heterozygous individuals presented as a double band (bisalbuminaemia or alloalbuminaemia) or, less frequently, as a single widened band. On the other hand, the rare homozygous cases presented as a net single band [15]. This technique allows to classify the variants into two main groups, on the basis of their migration speed: albumins that migrate faster (more anodic) with respect to normal ALB, and those that migrate more slowly. However, standard serum protein electrophoresis allowed to detect only alloalbumins with mutations involving charged residues placed on the surface of the molecule. In recent years, the introduction of more sophisticated techniques, such as electrospray time-of-flight mass spectrometry, identified new variants, with little or no differences in charge when compared to the normal protein (silent albumins, as ALB Lyon, ALB South Pacific, ALB Ilan). Nevertheless, there are probably many alloalbumins not yet detected thereby underestimating the frequency of the variants in the general population [15]. The values of 1:1000–1:3000 [1] or 1:300–1:400, as reported by Neel et al., using a more sensitive electrophoretic methodology [21], are, therefore, to be considered as merely indicative. In an isolated, minimally admixed cohort, the frequency can be much higher. For example, ALB Ortonovo was only found in people from two small villages in Liguria, Italy, with a frequency of ≥1% [22], and the incidence of alloalbumins in the Kannikar tribes in Southern India and in some American Indian Tribes were reported to be 9.5% and ≥20%, respectively [15].

### 2.2. DNA and Protein Changes

All the 14 translated exons, as well as all the regions encompassing the 5′ end of exon 14 and the 3′ end of the following intron, are involved in determining the 74 albumin variations. On the other hand, most of the amino acid changes are clustered in three regions of the protein. Of these, one is located in the propeptide and two in the amino terminus (residues 313–382 corresponding approximately to subdomain IIB in the crystal structure, and residues 501–575 of the mature protein corresponding to subdomain IIIB). A lower frequency of incidence in mutations was reported in the homologous subdomains IIA and IIIA, each of which contains one of the hydrophobic cavities that are the principal binding sites for ligands [23,24]. This distribution suggests that the frequency of alloalbumins is probably not related to the variation sites in the gene but rather to the position of the mutated residue in the protein. The exposure of the mutated residues to the solvent may represent a key point: as mentioned above, alloalbumins were mostly detected by electrophoretic analyses, therefore, variations affecting the binding sites resulted in them not being easily detectable because of their location in hydrophobic areas poorly accessible to the solvent.

### 2.3. Pro-Albumin Variants

Pro-albumin is the ALB precursor with the propeptide, (RGVFRR) bound to the N-terminus. The numbering of these residues was traditionally from −6 to −1 (the juxtaposition to ALB itself), and those of the mature ALB molecule from 1 to 585. The addition of 24 to these numbers converts them to a number according to HGVS rules, so the current numbers are from 19 to 24. In a healthy condition, pro-albumin is cleaved by a specific propeptidase within the hepatocytes before the mature protein is released in blood-stream, therefore, it is not detectable in normal serum. The propeptidase (convertase) is a diarginyl-specific, Ca^++^-dependent membrane bound endopeptidase [25].

However, in pathological conditions, such as viral acute hepatitis caused by the delta virus, or inhibition of the propeptidase due to mutant forms of α1-antitrypsin, pro-albumin may be detected in circulation [15]. Apparently, no amino acid substitutions in positions between −6 and −2 interfere with this post-translational process. In contrast, the substitution of Arg -2 or Arg -1 inhibits the proteolytic cleavage of the pro-peptide but not the serum release of the pro-protein. The six variants today known and representing the first identified cluster are Malmö I, –2Arg → Cys (p.Arg23Cys); Lille, –2Arg → His (p.Arg23His); Christchurch, –1Arg → Gln (p.Arg24Gln); Takefu, –1Arg → Pro (p.Arg24Pro); Jaffna, –1Arg → Leu (p.Arg24Leu); Blenheim, 1Asp → Val (p.Asp25Val). These findings show that the substitution of only one of the two arginines in position −1 or −2, as well as the substitution 1Asp → Val, involving the N-terminal residue of the mature ALB molecule, impair the final maturation in hepatocytes. Of note, the first five variants are due to mutations in the codon CGA for Arg-1 (Christchurch, Takefu, Jaffna) and CGT for Arg-2 (Malmö I, Lille). The codon for cytosine residues contain the prone CpG sequences in either the sense or the antisense strands, giving rise to CpG, TpG, or CpA transitions. Mutations in the sense strand are Malmö I, c.67C>T, whereas Lille, c.68G>A, and Christchurch, c.71G>A, are CpG to CpA transitions on the sense strand, with the initial variation presumably occurring in the antisense strand. As reported in Table 1, the Blenheim, 1Asp → Val mutation may be present in populations that are genetically distinct and from different geographical areas [26]. Additionally, the Malmö I mutation, although present only in Caucasians, has been found in families of three different European nationalities (Swedish, Scottish, and Italian), with a frequency of approximately 1:1000 [26].

ALB binds Cu^++^ and Ni^++^ at its N-terminal DAH sequence [1]. Thus, it is not surprising that modifications involving this site causes low binding for these two metal ions. This hypothesis was confirmed by spectrophotometric and equilibrium dialysis studies, which evidenced total suppression of the binding of Cu^++^ to pro-ALB Christchurch and the almost complete absence of high-affinity Ni^++^ binding to pro-ALB Lille, Christchurch, and Blenheim and to Arg-ALB, whereas the association constant for binding to ALB Blenheim, which has no additional residues at the N-terminus, was halved [15]. This pronounced effect on binding could have an impact on the biological fate of the two metal ions. On the other hand, the presence of the propeptide has only minor effects on Ca^++^ and Zn^++^ binding, which takes place elsewhere in the protein [15].

### 2.4. Mutations in the ALB May Involve Hypermutable CpG Dinucleotides

The presence of hypermutable CpG sequences in ALB may explain the high frequency of mutations in the pro-peptide encoding sequence [26], as well as the presence of hot spots in the mature molecule [27], or the detection of the same mutation in unrelated analbuminaemic individuals [28]. These mutations occur in cytosine residues in CpG sequences in either the sense or the antisense strand, giving rise to CpG, TpG, or CpA transitions. Whether the higher frequency of mutations identified in the diarginyl sequence is a marker of hypermutability or is the result of experimental bias has not yet been clarified. Variants of pro-albumin are easy to detect in conventional SPE because of their relative stability, the exposure to the solvent of the modified propetide, and their slow mobility due to the +2 charge, which results in discrete bands. In contrast with the up to five-point mutations identified in the codons CGA and CGT modifying −1 Arg and −2 Arg in the ALB propeptide, only a few point mutations involving CpG dinucleotides have been identified in the exons coding for the 585-residue mature of albumin molecule. A significant example is the c.1552G>A variation, which was identified in albumins Casebrook and Besana Brianza, suggesting that it may represent a hot-spot in *ALB* [29]. In addition, a G to A transition in the second nucleotide of the codon for Arg218 (c.725G>A) in a CpG sequence is the molecular basis of the Arg218His mutation, which is by far the most common cause of FDH-T4 [30]. This mutation was found in many different countries and ethnicities (Appendix A Appendix A), suggesting that the CpG sequence in the codon for Arg218 also represents a hot-spot in *ALB*. Additionally, the c.724C>T (Arg218Cys) mutation, reported in the Exome Aggregation Consortium Website [31], occurs in this CpG sequence but it does not represent a cause of FDH-T4.

### 2.5. N-Glycosylated Variants

Albumin is normally unglycosylated because its primary structure lacks the canonical Asn-Xaa- Thr/Ser tripeptide acceptor sequence required for N-glycosylation. However, 20 different point mutations in the albumin gene could theoretically generate this sequence [15,20]. To date, three such point mutations have been identified in ALB variants and all tripeptide acceptor sequences are N-glycosylated: ALB Dalakarlia 63Asp → Asn (Asp87Asn); ALB Redhill −2Arg → Cys (Malmö-I); 320 Ala → Thr (p.Arg23Cys; p.Ala344Thr); ALB Casebrook 494 Asp → Asn (p.Asp518Asn) [15,20]. The tripeptide acceptor sequence originated by the variations are Asn-Lys-Ser for ALB Dalakarlia, which is N-glycosylated at 63Asn; Asn-Tyr-Thr for ALB Redhill, which is N-glycosylated at 318Asn; Asn-Glu-Thr for ALB Casebrook, which is N-glycosylated at 494Asn [20]. In all the reported cases, the glycan is a disialylated (mainly or totally) biantennary complex-type oligosaccharide; however, the variants exhibited a different degree of glycosylation [15,32,33].

ALB Redhill is one of the two variants characterized by the presence of two independent mutations in the same allele. In addition to the 320 Ala → Thr, which leads to N-glycosylation of 318Asn (see above), the other is −2Arg → Cys as for proALB Malmö-I, which, in this case, results in abnormal hydrolysis of pre-pro-albumin within the liver cells and to the formation of ALB still possessing an Arg at position −2 [20,26]. The other alloalbumin with two independent variations is ALB South Pacific 540 Thr → Ala (p564ThrAla), 546 Ala → Ser (p570Ala Ser). The two silent substitutions, without charge difference with respect to normal ALB could be identified by the use of a more sensitive technique such as electrospray time-of-flight mass spectrometry of whole plasma [20,34]. The variant ALB was present at a lower level with an expression ratio of approximately 1:2 (variant vs. normal).

### 2.6. C-Terminal Variants

More extensive alterations of the molecule were observed in six C-terminal variants: five among them (Bazzano, Catania, Rugby Park, Banks Peninsula, and Venezia) were truncated, and one (Kénitra) showed an extended polypeptide chain [20]. The single nucleotide deletion found in ALB Bazzano (p.Cys591Alafs*17) is located in exon 13 (c.1771delT) and involves the first base of the codon for 567Cys. The subsequent frame-shift originates a divergent amino acid sequence spanning 16 residues, beginning at position 567, with an early termination codon at position 582. The C-terminal sequence is: ALPRRVKNLLLQVKLP. It should be noted that the mutation of albumin Bazzano has caused the loss of the C-terminal disulfide bridge. The variant represents approximately 18% of the total albumin in heterozygous subjects [35]. The other variant caused by a single nucleotide deletion and by the subsequent frame-shift is ALB Catania (p.Gln604Lysfs*4). In this case, the deletion of the cytosine residue at the first position of the codon for Gln580 (c.1810delC) leads to a frame-shift mutation in exon 14, generating a shortened C-terminus sequence because of the introduction of a termination codon at position 583. Residues 580–585 (QAALGL) are replaced by residues 580–582 (KLP). The variant represents approximately 30% of the circulating protein in heterozygous subjects [36,37].

3′ splice site mutations generate ALB Rugby Park, ALB Banks Peninsula, and ALB Venezia. ALB Rugby Park (p.Gly596_Leu609delinsLeuLeuGluPheSerSerPhe) is caused by a single point substitution (c.1785+1G>C) involving the first base of a donor splice site: this G to C mutation at position 1 of the 13th intron, with the replacement of the obligate GT sequence by CT at the exon/intron boundary, prevented splicing of the 13^th^ intron and translation continued for 21 nucleotides until a stop codon was reached. The variant protein, which represents approximately 8% of total albumin in heterozygous subjects, lacks the 14 amino acids coded for in the 14th exon (GKKLVAASQAALGL), but these are replaced by 7 new residues (LLQFSSF), giving a truncated molecule of 578 residues [38]. ALB Banks Peninsula (p.Gly596_Leu609delinsSerLeuCysSerGly), a genetic variant that is expressed at only 2% of the total serum albumin content, is caused by a single T to A point mutation in intron 13 (c.1786-15T>A), 15 bases upstream of the intron 13/exon 14 boundary. This mutation creates a new AG dinucleotide, the invariant sequence encountered in all eukaryotic intron acceptor splice sites. This splicing mutation results in the translation of the 3′ terminal region of intron 13 and of the 5′ region of exon 14 as SLCSG before a TAA terminator is reached. Predictably, the polypeptide chain consists of 576 residues with the replacement of the C-terminal GKKLVAASQAALGL sequence by SLCSG [39]. A more extensive DNA mutation (c.1786_1814+1delinsAAAAT) was observed in ALB Venezia (p.Gly596Profs*10). The 30-bp deletion and the 5-bp insertion alters the first consensus nucleotide of the donor splice junction of intron 14 and the 3′ end of exon 14 which is shortened from 68 to 43 bp. This change leads to an exon skipping event resulting in the direct splicing of exon 13 to exon 15. The protein product has a truncated amino acid sequence (578 residues instead of 585), and the C terminal is altered after Gly572: PTMRIRERK. The variant C-terminal end with the dibasic sequence Arg-Lys is removed only partially, following stepwise cleavage in the circulation by basic carboxypeptidases to yield different forms of the circulating protein [37,40].

A single A duplication in exon 14 (c.1794dupA), and a subsequent frame-shift, cause ALB Kènitra (p.Leu599Thrfs*30), which represents approximately 15% of the total plasma ALB in two members of a family of Sephardic Jews from Kènitra (Morocco). The frame-shift caused the translation to the first termination codon of exon 15, which normally does not code for the protein but is conserved in the mature mRNA. The resulting amino-acid sequence is completely variant from residue 575 to the C-terminal end: TCCCKSSCLRLITSHLKASQPTMRIRERK. The seven C-terminal residues of ALB Kènitra correspond to the first seven codons of exon 15, while the 8th and the 9th codons specify arginine and lysine, respectively, and are followed by a terminator codon (TGA). No evidence was found for the presence in the circulating variant of the C-terminal dipeptide Arg-Lys, which is probably cleaved off in the circulation by basic carboxypeptidases. The four additional cysteine residues form two new S–S bridges, and Thr596 is partially O-glycosylated by a monosialylated oligosaccharide, the glycosylated form representing approximately 50% of the variant. Thus, ALB Kénitra is peculiar because it represents the only case known so far of an elongated (601 residues instead of 585) and O-glycosylated mutant of ALB [41].

### 2.7. In Vivo Stability

As seen above, the serum level of the six C-terminal truncated or elongated variants ranges from 2 to 30% of the total ALB in heterozygous subjects, suggesting that the COOH-terminal end is fundamental for ALB stability [36,40]. Andersen and co-workers showed that the reduced serum concentrations of variants modified in the C-terminal is due to the lower recycling [42,43]. Normally, after cellular uptake, part of ALB is bound to the intracellular neonatal Fc receptor FcRn. This binding limits intracellular proteases and ALB is partially released again in the circulation. A further study based on the comprehensive use of recombinant ALB variants revealed that His488, His534, and His559 are crucial for reversible and pH-dependent binding of ALB to the receptor [44]. Therefore, the ALB C-terminal is fundamental to maintain the plasmatic lifetime and may be the target of future therapeutical approaches with the aim to increase the availability of liganded ALB drugs [45]. The structure of domain III is crucial for the pH-dependent ALB binding to FcRn, and mutations in this domain could reduce binding and shorten molecular survival, as reported for the glycosylated variant ALB Casebook [15,20]. On the other hand, different mutations in domain III could strongly stabilize the protein, resulting in a longer lifetime [45].

As already reported, the synthesis of ALB is regulated by a single-copy gene co-dominantly expressed, and mutations in heterozygous individuals usually result in the co-presence of the normal and the variant proteins in a 1:1 ratio. However, in addition to the six C-terminal mutants, other alloalbumins are present in serum with lower incidence. Among others, Alb Larino 3His → Tyr (p.His27Tyr) represents only 10–12% of the total ALB: the mutated base is the last nucleotide of exon 1 and, consequently, its substitution could alter splicing, resulting in low expression of such variants [35]. ALB Hawkes Bay 177Cys → Phe (p.Cys201Phe) represents only 5% of total circulating ALB. Its low plasma level is probably due to molecular instability caused by the loss of the sulfhydryl bridge between 168Cys and 177 Cys and a rearrangement of one of the pre-existing S–S bridges [46]. An alteration of the normal pattern of the disulphide bonds is likely also the cause of the low serum level (20–25%) of the variant ALB Asola 140Tyr → Cys (p.Tyr164Cy). Despite the presence of an additional cysteine residue, several lines of evidence indicated that this alloalbumin has no free -SH group, probably because of the formation of a new S-S bond between Cys140 and Cys34, the only free sulphydryl group present in the normal protein, and of the subsequent gross conformational change of the molecule [47].

Even more unusual is the behavior of ALB Caserta 276 Lys → Asn (p.Lys300Asn), the amount of which was from 60–65% of the total albumin, suggesting an increased stability of the variant. Lys276 is an exposed residue that may play an important role in the chemical modification of ALB during its degradation [35].

### 2.8. Familial Dysalbuminaemic Hyperthyroxinaemia (FDH-T4) and Hypertriiodothyroninaemia (FDH-T3)

Mutations generally have limited effect on ligand binding with limited clinical significance. However, the mutations 218Arg → His, 218Arg → Pro, 218Arg → Ser, and 222Arg → Ile are reported in familial dysalbuminemic hyperthyroxinaemia (FDH-T4,OMIM # 615999), and 66Leu → Pro in familial dysalbuminemic hypertriiodothyroninaemia (FDH-T3) [30]. FDH-T4 and FDH-T3 are dominantly inherited syndromes characterized by a high concentration of thyroid hormone in the blood-stream. The syndromes do not cause disease because the concentration of free hormones is normal [30]. However, these variants are not detectable upon electrophoresis, they were discovered by the large increase of T3 and T4 in serum. This finding was confusing to endocrinologists until the causative variations in the gene were established. To avoid inappropriate surgical or drug treatment of euthyroid subjects with hyperthyroxinaemia or hypertriiodothyroninaemia, protein and/or DNA sequencing of the ALB should be performed [30]. FDH-T4 is the most common cause of euthyroid hyperthyroxinaemia in Caucasian populations, with prevalence of around 1 in 10,000 individuals, with higher frequency in some specific ethnic groups [30]. The disorder is characterized by greater elevation in serum L-thyroxine (T4) than in serum triiodothyronine (T3). The high serum concentration of T4 is due to the modification of a binding site located in the N-terminal half of ALB (in subdomain IIA). Thus, mutating Arg218 or Arg222 for a smaller amino acid reduces the steric restrictions in the site and creates a high-affinity binding site. The mutations can also affect the binding of other ligands and can perhaps cause modified pharmacokinetics of ALB binding drugs. In normal ALB, the high-affinity site has another location (in subdomain IIIB) [48]. Different locations of these sites imply that persons with and without FDH-T4 can have different types of interactions, and thereby complications, when ALB-binding drugs are administered. FDH-T3 is caused by a leucine to proline mutation in position 66 of HSA, resulting in a large increment of the binding affinity for T3 but not for T4. Site-directed mutagenesis in position 242 could be therapeutically relevant for the treatment of hyperthyroidism, which is a disease state in which free serum L-thyroxine is increased, e.g., in cases of thyrotoxicosis or thyroid storm, if the mutant is administered intravenously then it could bind L-thyroxine and thereby rapidly lower the concentration of the free, biological active hormone. Such treatment would be relevant, especially for pregnant patients because ALB does not cross the placental barrier [15].

### 2.9. Fatty Acids, NO, and Hydrophobic Drugs Binding to the Variants

As seen above, some alloalbumins have pronounced effects on the binding of Ni^++^, Cu^++^, and thyroid hormones; however, ALB can bind many other ligands. Genetic variants with known mutations were used to obtain valuable molecular information on the binding sites of the protein. The main results are reported in the Appendix A, Appendix A. Such information can be further clarified by the use of site-directed mutants. Here, we discuss the results obtained with the other most relevant different ligands.

Long-chain fatty acids represent the most relevant metabolites transported by ALB. They are rapidly metabolized, having a half-life in plasma of only about 10 min, and 30–40% are removed in 5–10 s by the liver [1]. Gas chromatographic studies on the fatty acid load of 30 pro-albumin and ALB variants, and of their corresponding ALB A isolated from heterozygous individuals, showed that in some cases the total amount bound to the variant was reduced when compared to the amount bound to ALB A. In seven cases the variants contained >50% more fatty acids than normal protein (Appendix A Appendix A). The most pronounced changes and the majority of cases of increased binding were caused by molecular changes in domain III (Appendix A Appendix A).

Several dialysis techniques were used to study the high-affinity binding of laurate to defatted preparations of 17 variants, which showed only minor effects, probably because the natural mutations are usually situated on the surface of the protein [15]. The binding of octanoate, decanoate, laurate, and myristate to 13 recombinant isoforms mutated in four different binding regions was studied by X-ray crystallography [15]. All the recombinant variants showed a pronounced diminishing effect, directly or via conformational changes, on fatty acid binding. The study also revealed that, in addition to binding affinity, the number of high-affinity sites and their locations in the protein depend on the chain length of the fatty acid.

The S-nitrosylated form of ALB seems to be a major circulating endogenous reservoir of nitric oxide (NO) and may have potential as a NO donor in therapeutical applications. The alloalbumin Liprizzi 410 Arg → Cys (p.Arg434Cys) can bind NO at both the normal Cys34 and at the new Cys410, the latter being more rapidly and efficiently nitrosylated than Cys 34 [49]. S-nitrosylated ALB Liprizzi exhibited strong antibacterial activity, the suppressed apoptosis of human promonocytic cells, and reduced liver damage in a rat liver ischemia–reperfusion injury model [49]. Perhaps the recombinant introduction of additional free sulfhydryl groups could increase this function even further, although, whereas NO transfer from ALB with only one or a very few NO molecules bound to it has cytoprotective effects, the transfer of large amounts of NO has cytotoxic effects [50]. On the other hand, the latter effect could be useful in cancer therapy [51].

Hydrophobic drugs, especially those with a negative charge, are frequently bound to ALB. Of the nine variants tested, two (ALB Tagliacozzo and ALB Parklands) showed very low affinities for diazepam, salicylate, and warfarin (Appendix A Appendix A) [15]. In the remaining 21 drug-variant combinations, binding was either normal or diminished to a less pronounced extent. Finally, recombinant versions of FDH-2 and FDH-3 have low binding affinity for warfarin (Appendix A Appendix A) [15].

### 2.10. Pharmacokinetics

The pharmacokinetics of 17 variants with single-residue substitutions, together with three glycosylated and three truncated variants, has been studied in rodent animal models [15]. The plasma half-lives for three of them were greatly reduced compared to the corresponding endogenous ALB (*p* < 0.01): ALBs Tregasio, variant Hawkes Bay mutated in domain I and ALB Venezia, a C-terminally truncated variant. Among the variants with single-residue modifications, the following general trends were found: only variants with +2 changes in the net charge had slightly prolonged half-lives, whereas changes in hydrophobicity decreased it. Furthermore, both positive and negative correlations were found between changes in α-helical content located in domains I + III and domain II, respectively, and changes in half-live of the modified ALB molecules. Iwao et al. reported that the recombinant mutant Arg434Ala has a very short plasma half-life [52] due to an increased cellular uptake by membrane receptors in the liver (the liver clearance was increased 27-fold) or other organs and/or to a decreased binding to the intracellular receptor FcRn. The short plasma half-lives of ALBs Tregasio, Hawkes Bay, and Venezia can partly be explained by increased liver uptake. The liver clearance of ALB Hawkes Bay was increased 20-fold when compared with normal protein, and ALBs Herborn, Caserta, Roma, Bazzano, Dalakarlia-1, and Casebrook also had an increased liver uptake. On the other hand, ALBs Parklands, Redhill, Maku, and Arg-ALB had reduced hepatic clearance. Finally, Nakajou et al. have produced a triple-mutant of ALB (Lys223Ala, Lys463Ala, Lys549Ala), and the substitution of the three lysines caused a 6-fold increase in liver clearance [53]. The examples of increased liver uptake could be useful for the development of selective drug targeting systems to the organ. Thus, Hirata et al. have made three recombinant alloalbumins with the substitutions Asp87Asn, Ala344Thr, or Asp518Asn as well as one mutant with all three aminoacid changes [54]. The four mutants have N-linked oligosaccharides composed of GluNAc and D-mannose. Pharmacokinetic studies with mice showed that the hepatic uptake clearance of the highly mannosylated Asp518Asn mutant and of the triple mutant were increased 16-fold and 47-fold, respectively, whereas the uptake of the less mannosylated Asp87Asn and Ala344Thr mutants were similar to that of ALB. Additional investigations with the triple-mutant strongly suggest that the increased uptake is due to selective interaction with mannose receptors on Kupffer cells. The potential use of the triple mutant for specific targeting to the liver of drugs and other ligands was supported by its cytoprotective delivery of nitric oxide to the Kupffer cells in a liver ischemia–reperfusion injury model. Increased kidney uptake clearance was observed for several variants, namely ALBs Hawkes Bay (6-fold), Caserta, Tagliacozzo, Maku, Milano Fast, Bazzano, Venezia and Catania. The only isoform with a decreased uptake was ALB Redhill [55,56]. The altered clearances are most probably caused by a modified glomerular filtration and/or modified interaction with the renal tubular membrane receptors responsible for reabsorption of ALB. A further explanation may be a modified kidney uptake by tubular receptors for advanced glycation end products (RAGE). Uptake by the spleen has only been examined for the three glycosylated variants and the three C-terminally truncated variants [57]. Among these, the clearance of ALB Bazzano was increased, while spleen clearance of ALBs Dalakarlia-1 and Redhill was decreased. The molecular explanation for these effects and the type of membrane receptor(s) involved are still unknown.

### 2.11. Future Perspectives

ALB variants are not associated with disease. However, the detection of alloalbumins may represent an innovative tool to track millennial human migration, providing possible models for the study of neutral molecular evolution. As an example, ALB Naskapi has been found with relevant frequency in the Algonquian speaking people from the East Coast of United States and Canada [58]. The variant is structurally equal to the ALB Mersin variant, detected with variable frequency in Eti Turks from southeastern Turkey [59]. These findings may suggest a genetic relationship between Eti Turks and American Indians [59]. Genetic variants with known mutations can also provide valuable molecular information about ALB binding sites, antioxidant and enzymatic properties, as well as stability (see above). A better understanding of ALB binding sites, with the innovative aim of genetic manipulation, may be useful in different clinical contexts: in regulating the clearance of fatty acids, as previously mentioned, through mutation at position 218 or for the presence of additional SH group(s); high-affinity mutants may have an application in the emergency serological accumulation of endogenous substances, such as drug overdose; in dialysis, which is an extracorporeal treatment for patients with end stage renal disease consisting in the removing of water-soluble and ALB-bound toxins [57].

## 3. Congenital Analbuminaemia (CAA)

### 3.1. Diagnosis, Frequency, and Symptoms

Hypoalbuminemia is an acquired condition, defined by serum levels of ALB level < 3.5 g/dL, due to the decreased production in the liver or to the increased loss in the gastrointestinal tract or kidneys, increased use in the body, or abnormal distribution between body compartments. Patients often present with hypoalbuminaemia as a result of several diseases such as sepsis, hepatic cirrhosis, nephrotic syndrome, or protein-losing enteropathy. Moreover, hypoalbuminaemia in hospitalized patients is a strong predictor marker of mortality and morbidity [60,61]. In contrast, congenital analbuminaemia (CAA; OMIM # 616000) is an autosomal recessive disorder characterized by the absence, or a low level, of ALB in serum, caused by variations in the ALB [18]. Many methods are commonly used in clinical chemistry laboratories to establish the ALB concentration, such as conventional or capillary serum protein electrophoresis, standard clinical chemistry systems using photometric dye-binding, and immunochemistry techniques [62]. Of these, dye-binding assays overestimate ALB levels at low concentrations [62] and concentrations around or greater than 15 g/L have recently been reported in analbuminaemic subjects [18]. This, together with the absence of unambiguous clinical and biochemical signs in the analbuminaemic individuals (see below), may represent difficulties in the early clinical diagnosis of CAA. In contrast, immunonephelometric techniques in association with serum protein electrophoresis represents the most accurate method to test a well-founded suspicion of CAA, being able to detect serum ALB levels < 1 g/L [18]. In conclusion, considering the technical difficulties in determining the right ALB values in the above reported clinical conditions, a genetic test is mandatory to confirm the diagnosis of CAA. It is a very rare condition, with a prevalence of less than 1:1,000,000, without gender or ethnicity prevalence [1]. After the first case of a 31-year-old German woman reported in 1954, only about 90 cases caused by 28 different mutations have so far been described world-wide [20]. The frequency of the trait can be much higher in restricted and minimally admixed population groups than in the average population, as reported in the First Nations Communities of Cree Descent living in the northwestern central plains regions in Saskatchewan (Canada), and in the residents of a Slovak gypsy settlement [18]. CAA is benign in adulthood because the compensatory increase of different plasma proteins compensate the lack of ALB functions, reducing the potential severity of the symptoms. Globulins represent the most abundant serum protein in the case of CAA; although, lipoproteins, transferrin, α1-antitrypsin, thyroxin binding globulin, and coagulation factors are also increased. As a clinical result, the total serum protein level is only slightly decreased and CAA subjects present moderate clinical symptoms such as mild oedema, hypotension, and fatigue [1,18]. Some adult patients, especially females, develop a peculiar lower body lipodystrophy, often with abnormal body habitus [1,18]. However, long-term follow-up reports are lacking. The Albumin website reports that one female died at the age of 32, whereas six other persons lived until they were 55–75 years old [20]. In contrast with the apparently mild symptoms in adulthood, CAA can have serious consequences during the prenatal period, causing miscarriages and preterm birth, and leading to death in early childhood, mainly due the fluid retention and infections of the lower respiratory tract [60]. The rarity of the condition may be explained considering that only a limited number of analbuminaemic individuals survive past the pre- and peri-natal period [18]. A confirmation of this hypothesis is provided by a recent survey, showing that CAA can have serious consequences in the pre- and peri-natal period [63]. The absence of the protein in the fetus is an important cause of preterm birth [63]. In addition, few reports described the death of several CAA individuals in early infancy [18]. CAA has been reported as being the second most common direct cause of death in children younger than 5 years [63].

### 3.2. Molecular Genetics

Of the 90 CAA cases recorded, 56 were studied at the molecular level, allowing the identification of 28 different causative defects. The results are summarized in Table 2. In all the cases, except for case #32 of the Registry (www.albumin.org accessed on 15 October 2021), mutations were found in the first 12 exons of the albumin gene and in the intron/exon junctions.

The results show that CAA is an autosomal recessive disorder caused by the inheritance of abnormal ALB alleles from both parents. In the heterozygous state, the single normal allele is sufficient to produce approximately half of the normal amount of ALB, and individuals generally have ALB concentrations close to the lower limit of the normal range (approximately 30–35 g/L).

Of the 28 different mutations, 26 were identified in analbuminaemic subjects as being homozygous for the trait. They include 1 mutation in the start codon (Afula, c.1A>C), 2 frame-shift/insertions (Roma, c.872dupA, and Gazaoueth, c.1098dup), 6 frame-shift/deletions (Kayseri, c.228_229delAT, Amasya, c.229_230delTG, Erzurum, c.527delC, Bologna c.920delT, Locust Valley, c.1610delT, and Safranbolu, c.1610delT), 7 nonsense variants (Codogno, c.166C>T, Bethesda, c.412C>T, Seattle, c.714G>A, El Jadida, c.802G>T, Monastir, c.1275C>A, Hama, c.1309A>T, and Tubingen, c.1525C>T), and 10 mutations affecting splicing. (Baghdad, c.79+1G>A, Madeira, C.138-2A>G, Treves, c.270+1G>T, Zonguldak, c.597T>A, Nijmegen-2, c.615G>A, Vancouver, c.714-2A>G, Guimarães, c.1289+1G>A, Tripoli, c.1428+1 G>T, Bartin, c.1428+2T>C, and Ankara, c.1652+1G>A). Compound heterozygosity for the remaining two molecular defects, a nonsense mutation (Roma-2, c.1225C>T) and a splice site mutation with a subsequent reading frame-shift (Fondi, c.1427A>G), caused CAA in an Italian man (case #23 of the Register). Thus, nonsense mutations, mutations affecting splicing, and frame-shift/deletions seem to be the most common causes of CAA.

The length of the abnormal polypeptide chains, for the twenty cases in which a prediction can be made, would range from 31 (Codogno) to 532 (Locust Valley) amino acids. However, no evidence has been found for the presence in serum of the putative protein products that originated as a consequence of the 28 mutations, since an intact C-terminal end of the molecule is required for its long plasma half-life [40].

The majority of the causative molecular defects are unique, detected in only a single individual or in members of the same family. Exceptions are Bethesda, El Jadida, Safranbolu, Guimarães, and especially Kayseri. Both the El Jadida and the Safranbolu mutations were shown to cause CAA in two subjects. The c.412C>T Bethesda mutation, which was identified in three unrelated individuals, lies in a CGA codon containing a hypermutable CpG dinucleotide site, and a different variation (c.412C>G; p.Arg138Gly) produces ALB Yanomama, an alloalbumin present in polymorphic (>1%) frequency in an Amazonian Indian tribe [20]. A relatively more common mutation is the splicing defect of analbuminaemia Guimarães (c.1289+1G>A), identified in five analbuminaemic individuals belonging to four unrelated families. To date, the AT deletion at nucleotide positions c. 228–229 of analbumianemia Kayseri is by far the most frequent cause of CAA identified, having been found in 14 individuals, belonging to geographically distant and apparently unrelated ethnic groups (Appendix A Appendix A). Therefore, it accounts for about one-third of the cases characterized at the molecular level. In addition, the frequency of this mutation seems to be significantly higher in restricted and minimally admixed population groups than that of CAA in the average population. Examples are two First Nation communities of Cree descent living in the northwestern central plains of Saskatchewan (Canada) (cases # 27, 28, 34, and 36 and footnote in the Register), which in recent years have decided to prevent members of their tribes from participating in further genetic studies, and a Slovak gypsy settlement (cases # 39-41), plus others currently under investigation by Dr. S. Rosipal (personal communication). Therefore, the Kayseri mutation likely accounts for about half of the known affected individuals world-wide [18,20].

## 4. Conclusions

Currently, the 28 different molecular defects within the *ALB* reported to cause CAA are located in ten different exons (1, 3, 4, 5, 7, 8, 9, 10, 11, and 12) and in seven different introns (1, 2, 3, 6, 10, 11, and 12). No cases were found in the last two coding exons (13 and 14) because such variations there would probably cause the presence of a circulating C-terminal variant of the protein and not CAA. In summary, the first twelve exons of the ALB gene, with the exception of the two shortest, exons 2 and 6, were reported to contain at least one molecular defect resulting in CAA [20,62], and twenty variations are located within those ten exons, whereas the remaining eight variant sites were identified within seven introns, suggesting that CAA is the result of a widely scattered random sequence of variations [64,65]. However, the increasing knowledge of the causative defects seems to bring to light the presence of regions in the *ALB* that are prone to mutations, as suggested by the presence of the same defect in unrelated analbuminaemic individuals. Two of them appear to be localized in the intron 6/exon 7 junction (Vancouver and Seattle) and in the exon 11/intron 11 junction (Fondi, Tripoli, and Bartin). Other hypermutable regions seem to be the sequence c.228–230 of exon 3 (Kayseri and Amasya), the sequence c.1610–1615 near the 3′ end of exon 12 (Locust Valley and Safranbolu), and the CpG sequence at position c.412–413 in the codon CGA for p.Arg138 in exon 4.

Only in one case out of the fifty-three characterized at the molecular level, case #32 of the Register of Analbuminaemia Cases [20], could the causative mutations not be found within the ALB gene (Minchiotti, personal communication, confidential). Therefore, the possibility that the absence of ALB may be caused by mutations in deep intronic regions of the gene, in remote regulatory elements, or by defects in the ALB binding region of the intracellular neonatal Fc receptor, FcRn, which prevents ALB from degradation in the lysosomes (see above), cannot be ruled out. Consanguinity was shown to be a factor in most cases in which it was possible to define the genealogical tree of the affected family.

A better understanding of the phenotype and the molecular genetics of CAA, a condition which still has negative characteristics, will require the identification of the molecular defects underlying new cases and an accurate follow-up of the known analbuminaemic subjects.

## Figures and Tables

**Figure 1 ijms-23-01159-f001:**
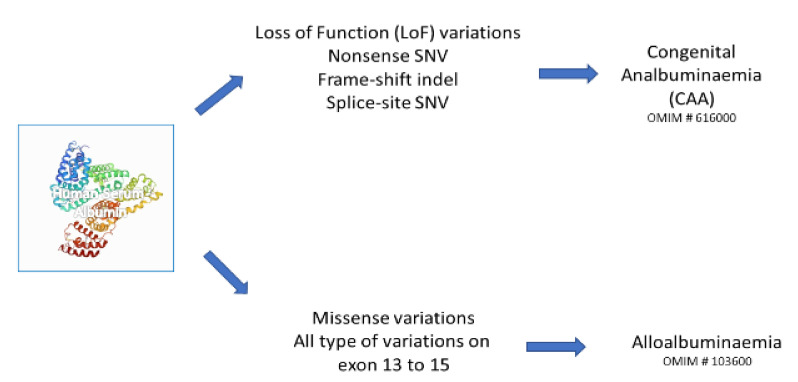
Effects of variants on albumin gene and their clinical consequences.

**Table 1 ijms-23-01159-t001:** Genetic variants of human serum albumin ^a^.

#	Name	Nucleotide and	Functional Effects
		Aminoacid Change ^(a)^	
1	Malmö I	c.67C>T p.(Arg23Cys)	Malmö I: 3% pro-albumin, 30% Arg-ALB (due to aberrant signal peptide cleavage) with the rest as normal albumin (ALB A). Sweden: 3 homozygotes.
2	Lille	c.68G>A p.(Arg23His)	
3	Christchurch	c.71G>A p.(Arg24Gln)	
4	Takefu	c.71G>C p.(Arg24Pro)	
5	Jaffna	c.71G>T p.(Arg24Leu)	
6	Blenheim	c.74A>T p.(Asp25Val)	Blenheim: 10% pro-albumin, 38% Val-. Bremen: 20% Arg-ALB, 30% Val-. In both cases, the remaining is ALB A. Blenheim: decreased α-helical content.
7	Larino	c.79C>T p.(His27Tyr)	Low in vivo stability.
8	Nagasaki-3	c.81C>A/G p.(His27Gln)	
9	Torino	c.250G>A p.(Glu84Lys)	
10	Dalakarlia-1	c.259G>A p.(Asp87Asn)	CHO next to a Cys. High thermal stability. Decreased α-helical content.
11	FDH-T3	c.269T>C p.(Leu90Pro)	High T3 binding. Identified in a Thai family.
12	Vibo Valentia	c.316G>A p.(Glu106Lys)	
13	Yanomama-2	c.412C>G p.(Arg138Gly)	Low bilirubin binding.
14	Nagoya	c.427G>A p.(Glu143Lys)	
15	Tregasio	c.437T>A p.(Val146Glu)	Decreased plasma half-life.
16	Komagome-2	c.455A>G p.(His152Arg)	
17	Asola	c.491A>G p.(Tyr164Cys)	20–25% variant.
18	Korea	c.593A>T p.(Lys198Ile)	
19	Hawkes Bay	c.602G>T p.(Cys201Phe)	5% variant. Decreased α-helical content. Decreased plasma half-life.
20	Ilam	c.643G>A p.(Ala215Thr)	Identified by a new high-resolution on-line reverse phase time-of-flight mass spectrometry procedure.
21	FDH-3	c.724C>A p.(Arg242Ser)	High free T4 andT3. Identified in a Canadian family of Bangladeshi extraction.
22	FDH-1	c.725G>A p.(Arg242His)	High T4 binding. Low warfarin binding. The most common causal variant in Caucasians.
23	FDH-2	c.725G>C p.(Arg242Pro)	High T4 binding. Low warfarin binding. Identified in Japanese and Swiss subjects.
24	FDH-4	c.737G>T p.(Arg246Ile)	High T4 binding. Identified in three unrelated African (Somali) families and one East European (Croatian) family.
25	Tradate-2	c.745A>C p.(Lys249Gln)	
26	Herborn	c.790A>G p.(Lys264Glu)	
27	Skaane	c.875A>G p.(Gln292Arg)	
28	Niigata	c.878A>G p.(Asp293Gly)	High prostaglandin binding.
29	Caserta	c.900G>C p.(Lys300Asn)	60–70% variant
30	Tagliacozzo	c.1011G>T p.(Lys337Asn)	Low drug binding. Low thermal stability. High progesterone binding.
31	Bergamo	c.1013A>G p.(Asp338Gly)	
32	Brest	c.1013A>T p.(Asp338Val)	High fatty acid binding.
33	Orebro	c.1026C>G p.(Asn342Lys)	
34	Redhill	c.[67C>T+1030G>A] p.[(Arg23Cys)+(Ala344Thr)]	High fatty acid binding.
35	Roma	c.1033G>A p.(Glu345Lys)	Low testosterone binding.
36	Sondrio	c.1069G>A p.(Glu357Lys	
37	Hiroshima-1	c.1132G>A p.(Glu378Lys)	
38	Coari I	c.1144G>A p.(Glu382Lys	
39	Trieste	c.1149G>T/C p.(Lys383Asn)	Low thermal stability.
40	Parklands	c.1165G>C p.(Asp389His)	Low drug binding.
41	Iowa City-1	c.1166A>T p.(Asp389Val)	
42	Benkovac	c.1175A>G p.(Glu392Gly)	
43	Naskapi	c.1186A>G p.(Lys396Glu)	
44	Nagasaki-2	c.1195G>A p.(Asp399Asn)	
45	Milano slow	c.1195G>C p.(Asp399His)	
46	Tochigi	c.1198G>A p.(Glu400Lys)	
47	Malmo-3	c.1198G>C p.(Glu400Gln)	
48	Hiroshima-2	c.1216G>A p.(Glu406Lys)	
49	Liprizzi	c.1300C>T p.(Arg434Cys)	High S-nitrosylation.
50	Dublin	c.1507G>A p.(Glu503Lys)	
51	Casebrook	c.1552G>A p.(Asp518Asn)	High fatty acid binding. Decreased α-helical content.
52	Vancouver	c.1573G>A p.(Glu525Lys)	
53	Ortonovo	c.1585G>A p.(Glu529Lys)	
54	Lyon	c.1601A>G p.(His534Arg)	
55	Maddaloni	c.1669G>A p.(Val557Met)	
56	Castel di Sangro	c.1678A>G p.(Lys560Glu)	
57	Wuxi	c.1684A>G p.(Lys562Glu)	
58	South Pacific	c.1690A>G p.(Thr564Ala)	
59	Maku, (Wapishana)	c.1693A>G p.(Lys565Glu)	High fatty acid binding. High thermal stability.
60	South Pacific	c.1708G>T p.(Ala570Ser)	
61	Mexico	c.1721A>G p.(Asp574Gly)	
62	Dalakarlia-2	c.1721A>C p.(Asp574Ala)	
63	Church Bay	c.1750A>G p.(Lys584Glu)	
64	Fukuoka-1	c.1759G>A p.(Asp587Asn)	High fatty acid binding. Decreased α-helical content.
65	Osaka-1	c.1765G>A p.(Glu589Lys)	
66	Bazzano	c.1771delT p.(Cys591Alafs*17)	High fatty acid binding. Decreased α-helical content.
67	B	c.1780G>A p.(Glu594Lys)	Low thermal stability.
68	Rugby Park	c.1785+1G>C p.(?)	8% variant. High fatty acid binding.
69	Banks Peninsula	c.1786-15T>A p.(?)	
70	Milano fast (Mi/Fg)	c.1789A>G p.(Lys597Glu)	This variant was recently identified in a 4-year-old Yemeni girl with growth hormone deficiency.
71	Vanves	c.1794A>T p.(Lys598Asn)	
72	Kénitra	c.1794dupA p.(Leu599Thrfs*30)	15% variant. Low thermal stability.
73	Catania (Ge/Ct)	c.1810delC p.(Gln604Lysfs*4)	
74	Venezia	c.1786_1814+1delinsAAAAT p.(?)	30% variant. Low thermal stability. Increased α-helical content Decreased plasma half-life

^(a)^ Nucleotide change accession number (NM_000477.7); Aminoacid change accension number (NP_000468.1); to view the complete Table 1 see Appendix A Appendix A.

**Table 2 ijms-23-01159-t002:** Variants causing analbuminemia in ALB gene.

#	Name	Nucleotide Change (NM_000477.7)	Aminoacid Change (NP_000468.1)
1	Afula	c.1A>C	Start-loss
2	Baghdad	c.79+1G>A	Splicing defect
3	Madeira	C.138-2A>G	Splicing defect
4	Codogno	c.166C>T	p.Gln56Ter
5	Kayseri	c.228_229delAT	p.Val78Cysfs*2
6	Amasya	c.229_230delTG	p.Val78Cysfs*2
7	Treves	c.270+1G>T	Splicing defect
8	Bethesda	c.412C>T	p.Arg138ter
9	Erzurum	c.527delC	p.Pro176Argfs*65
10	Zonguldak	c.597T>A	Splicing defect?
11	Nijmegen-2	c.615G>A	Splicing defect
12	Vancouver	c.714-2A>G	Splicing defect
13	Seattle	c.714G>A	p.Trp238Ter
14	El Jadida	c.802G>T	p.Glu268Ter
15	Roma	c.872dupA	p.Asn291Lysfs*8
16	Bologna	c.920delT	p.Leu307Argfs*23
17	Ghazaouet	c.1098dup	p.Val367fs*12
18	Roma-2	c.1225C>T	p.Gln409Ter
19	Monastir	c.1275C>A	p.Tyr425Ter
20	Guimarães	c.1289+1G>A	p.Phe398Alafs*33
21	Hama	c.1309A>T	p.Lys437Ter
22	Fondi	c.1427A>G	p.Tyr476Serfs*13
23	Tripoli	c.1428+1G>T	Splicing defect
24	Bartin	c.1428+2T>C	p.Leu431Tyrfs*5
25	Tubingen	c.1525C>T	p.Arg509Ter
26	Locust Valley	c.1610delT	p.Ile537Asnfs*21
27	Safranbolu	c.1614_15delCA	p.Leu540Phefs*2
28	Ankara	c.1652+1G>A	p.Leu477Cysfs*4

to view the complete Table 2 see Appendix A Appendix A.

## Data Availability

Data available on request from the authors.

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
