# Peer review of "Variations in the Human Serum Albumin Gene: Molecular and Functional Aspects"

_ijms, 2022, doi:10.3390/ijms23031159_

Round 1

Reviewer 1 Report

The review "Variations in the gene of human serum albumin: molecular and functional aspects and therapeutic possibilities" presents a comprehensive study about genetic and molecular aspects, functional consequences and potential therapeutic uses of the variants of albumine. Discussion about  molecular defects, stability, ligand binding and pharmacokinetics is included.

The review will be very useful for readers after its publication. 

I have only few minor - formal remarks:

  1. In the abstract, row 25: the dot at the sentence is missing.
  2. Figure 1, row 81: A figure capture is missing.
  3. In paragraph 3.1, row 463: morbidity[60,61] - a space is missing
  4. In paragraph 3.2, row 512: the additional row between a table and text is missing.

After correction these formal imperfections I recommend to accept the manuscript for publication in IJMS.

Author Response

I have only few minor - formal remarks:

  1. In the abstract, row 25: the dot at the sentence is missing. OK, Correct
  2. Figure 1, row 81: A figure capture is missing. Ok, we add the caption of Fig.1
  3. In paragraph 3.1, row 463: morbidity[60,61] - a space is missing. OK, Correct
  4. In paragraph 3.2, row 512: the additional row between a table and text is missing. OK, Correct

We corrected the formal/grammar imperfections evidenced by the  reviewer and performed a careful reading of the manuscript with the aim of making the English language used more correct and readable. We added a caption for Figure 1 and improved it reducing the size of the arrows.

Reviewer 2 Report

The authors presented the paper "Variations in the gene of human serum albumin: molecular and functional aspects and therapeutic possibilities"

1) The paper Fanali, et. al. Human serum albumin: From bench to bedside has some optimal information for the paper. The is the sections like Human serum albumin variants with altered fatty acid binding properties; Human serum albumin variants with altered metal binding properties; Human serum albumin variants with altered hormone binding properties which can be suitable for your Review.

2) Also, It will be very good to present in Table 1 functional effects like in Table S1.

3) In the paper name I see therapeutic possibilities, but I haven't seen enough discussion about such aspects as a new section. Albumin can be used for a lot of things for therapeutics production, for example. It will be very interesting to see the influence of the mutation and sub effects to the organism on using "wrong" albumin, or smth like this. 

4) It is known, that albumin is an allosteric protein. But I have seen any discussion of the mutation on binding sites efficiency or smth like this. Also, If you see the place of the mutation in some ways you will see, that the influence will be in another part of the protein. Of course, It will be very good to analyze the place of the mutation and its influence on the protein structure and binding sites, FcRn receptor binding, etc. But It will be hard work. That is why I just recommend presenting some of the examples may be as one more section.

5) Also, It will be good to do the picture of albumin and Its binding sites for various ligands, domains, amino acids numbers between the subdomains, N- and C-end, etc. It will be much easier to understand the mutation's place and the influence on the function and binding site.

Author Response

The authors presented the paper "Variations in the gene of human serum albumin: molecular and functional aspects and therapeutic possibilities"

1) The paper Fanali, et. al. Human serum albumin: From bench to bedside has some optimal information for the paper. The is the sections like Human serum albumin variants with altered fatty acid binding properties; Human serum albumin variants with altered metal binding properties; Human serum albumin variants with altered hormone binding properties which can be suitable for your Review.

We are perfectly aware that many reviews have already been published on the human serum albumin topic, many of which by members of our group. The paper "Human serum albumin: From bench to bedside 2012" is undoubtedly very thorough and comprehensive, but our choice was to write an updated mini-review that is more concise, but still able to provide the reader with all the essential information about genetic and molecular aspects, functional consequences and potential therapeutic uses of the variants (see reviewers  #1 and # 3). In particular, the material present in sections Human serum albumin variants with altered fatty acid binding properties; Human serum albumin variants with altered metal binding properties; Human serum albumin variants with altered hormone binding properties of the above mentioned paper was dealt with in our text with a different organization. The altered metal binding properties of the variants is discussed at the end of section 2.3. Pro-albumin variants, an altered thyroid hormones binding is reported in section 2.8. Familial dysalbuminaemic hyperthyroxinaemia (FDH-T4) and hypertriiodothyroninaemia (FDH-T3), and an extensive discussion about variants with altered fatty acid binding properties is present in section 2.9. Other ligands. In conclusion, we believe that the discussion of these points in our mini-review is sufficient, but we have deemed it appropriate to add a quote to the paper by Fanali et al. (ref # 4 in our revised version).

2) Also, It will be very good to present in Table 1 functional effects like in Table S1.

We decided to avoid "Functional effects" in the table 1 because if we added a column, we would have a  caracter size of the table too small and not readable.

3) In the paper name I see therapeutic possibilities, but I haven't seen enough discussion about such aspects as a new section. Albumin can be used for a lot of things for therapeutics production, for example. It will be very interesting to see the influence of the mutation and sub effects to the organism on using "wrong" albumin, or smth like this. 

Our mini-review is mainly focused on the molecular and functional aspects of the genetic variants,  but we believe that sections 2.10 and 2.11 can give to the reader enough information about potential therapeutic possibilities. As stated in point 1), the aim of this mini-review was not to reproduce the paper of Fanali et al. 

4) It is known, that albumin is an allosteric protein. But I have seen any discussion of the mutation on binding sites efficiency or smth like this. Also, If you see the place of the mutation in some ways you will see, that the influence will be in another part of the protein. Of course, It will be very good to analyze the place of the mutation and its influence on the protein structure and binding sites, FcRn receptor binding, etc. But It will be hard work. That is why I just recommend presenting some of the examples may be as one more section.

We agree that albumin is an allosteric protein . As an example,  we report in section in section 2.8 that the mutations responsible for FDH-T4 are located  in subdomain IIA, whereas in normal ALB, the high-affinity site for thyroxine has another location (in subdomain IIIB). Also, the influence of the mutation  on the protein structure and binding sites, as for FcRn receptor binding, is extensively discussed in our text

5) Also, It will be good to do the picture of albumin and Its binding sites for various ligands, domains, amino acids numbers between the subdomains, N- and C-end, etc. It will be much easier to understand the mutation's place and the influence on the function and binding site.

The figure requested by this reviewer has already been published by our group in the paper Kragh-Hansen, U., Minchiotti, L., Galliano, M., and Peters, T. (2013). Human serum albumin isoforms: genetic and molecular aspects and functional consequences. Biochim. Biophys. Acta 1830, 5405–5417. As already stated above, we cannot and do not intend to remake already published reviews.

Reviewer 3 Report

The Review entitled "Variations in the gene of human serum albumin: molecular and functional aspects and therapeutic possibilities" is a well-aimed paper for the summarization of the genetic aspects and the resulting therapeutic uses of the variants.

I suggest accepting the paper after careful rereading and correction of formal/grammar problems. Some examples of these are listed in the following.

Figure 1 has no caption and should be improved - too big arrows.

Shouldn't use dot at the end of the title, lines 3 and 437.

Line 25, missing dot at the end of the sentence. 

Author Response

I suggest accepting the paper after careful rereading and correction of formal/grammar problems. Some examples of these are listed in the following.

Figure 1 has no caption and should be improved - too big arrows. OK, Correct

Shouldn't use dot at the end of the title, lines 3 and 437. OK, Correct

Line 25, missing dot at the end of the sentence. OK, Correct

We corrected the formal/grammar imperfections evidenced by the  reviewer and performed a careful reading of the manuscript with the aim of making the English language used more correct and readable. We added a caption for Figure 1 and improved it reducing the size of the arrows.

Round 2

Reviewer 2 Report

1) About the Review Fanali, et. al. 

I mean that the names of your sectionÑ‹ may sound better and tell the reader comprehensive information. The name of the subsection like "other ligands" and some others does not show to the readers any information. 

Also, It was just my recommendation to use the information. There is some more interesting thing. It is for your consideration.

2) About Functional effects: It is much more interesting information than other columns. You can just do the one column with nucleotide and amino acid changes. Also, Protein change may be rewritten to the aminoacid change.

3) If you don't want to discuss the therapeutic possibilities, I highly recommend changing the title of the review and deleting "and therapeutic possibilities"

4-5) About allosteric properties and pictures. It was just a recommendation to improve the information in your review. It is for your consideration.

Author Response

Dear Editor,

Thank you for your fast revision, in attachment you’ll find the comments to the minor revision required by the Rev2.

As requested we are resubmitting the track changing paper only relative the current revision.

Reply to Rev2

1) About the Review Fanali, et. al. 

I mean that the names of your sectionÑ‹ may sound better and tell the reader comprehensive information. The name of the subsection like "other ligands" and some others does not show to the readers any information. 

Also, It was just my recommendation to use the information. There is some more interesting thing. It is for your consideration.

Thank you for your consideration, we agree with your suggestion and we changed the title of the 2.9 subsection from “Other ligands” to  “Fatty acids, NO, and hydrophobic drugs binding to the variants”; we hope that this could clarify better the paragraph content to the readers.

2) About Functional effects: It is much more interesting information than other columns. You can just do the one column with nucleotide and amino acid changes. Also, Protein change may be rewritten to the aminoacid change.

As you properly suggested we modified the table by joining two columns (nucleotide and amminoacid change) and by adding the “functional effects” column   

3) If you don't want to discuss the therapeutic possibilities, I highly recommend changing the title of the review and deleting "and therapeutic possibilities"

We agree and we change title

4-5) About allosteric properties and pictures. It was just a recommendation to improve the information in your review. It is for your consideration.

Thank you again for your suggestion, we believe it’s not necessary to present again already published data and figures also to preserve original work of all the groups working in the field. The reader can find every reference(s) and information in the text and in the bibliography.  

We hope that our manuscript in this revised form can qualify for publication in IJMS

With kind regards

Gianluca caridi & Lorenzo Minchiotti